# He–Ne Laser Priming Enhances Drought Tolerance in Wheat through Differential Modification of Photosynthetic Pigments and Antioxidative Enzymes

**Hamza Aslam [1], Muhammad Sajid Aqeel Ahmad [1,*], Ambreen Khadija Alvi [2], Wasifa Rani [1], Habib-ur-Rehman Athar [3], Ibrahim Al-Ashkar [4], Khalid F. Almutairi [4], Najeeb Ullah [5] and El-Sabagh Ayman [6,*]**

1   Department of Botany, University of Agriculture, Faisalabad 38040, Pakistan
2   Department of Botany, Government College Women University, Faisalabad 38000, Pakistan
3   Institute of Botany, Bahauddin Zakariya University, Multan 60800, Pakistan
4   Department of plant production, College of Food and Agriculture, King Saud University, Riyadh 11451, Saudi Arabia
5   Faculty of Science, Universiti Brunei Darussalam, Jalan Tungku Link, Gadong BE1410, Brunei
6   Department of Agronomy, Faculty of Agriculture, University of Kafrelsheikh, Kafr El Sheikh 33516, Egypt
*   Correspondence: sajidakeel@yahoo.com (M.S.A.A.); asabagh1980@yahoo.com (E.-S.A.)

**Abstract:** Water stress seriously impacts agro-systems worldwide, severely affecting plant growth and crop productivity. Radio-priming agents such as lasers can induce stress tolerance in plants due to their physiological roles in growth and development. The potential protective role of He–Ne laser pretreatment (i.e., one, two and five min at 630 nm) was evaluated in germination and growth of two wheat varieties, FSD-2008 and Anaj-2017, under water-stressed (50% field capacity) environments. Drought and laser priming significantly affected the growth (shoot and shoot fresh and dry weight and lengths), grain yield (number of total and fertile tillers and 100-grain weight), biochemical attributes (total soluble proteins and total free amino acids), and ionic concentration of both tested wheat varieties. In this study, the 2 min laser priming treatments were most effective for protecting wheat plants from drought stress. While the prolonged treatment duration significantly inhibited growth. We conclude that laser pretreatment assisted wheat plants in sustaining biomass assimilation, growth and yield formation by protecting their pigments and key metabolites from drought-induced oxidative injury. This study suggested that 2 min of laser priming had a much better result than other time duration, i.e., 1 and 5 min of laser priming.

**Keywords:** laser seed priming; exposure time; seed priming; water stress tolerance; wheat

## 1. Introduction

Current global warming and climatic anomalies are causing erratic rainfall patterns leading to prolonged and unpredictable drought spells all over the globe. Many regions of the earth are receiving lower-than-average rainfall per annum, resulting in periodic water shortages in agricultural soils [1]. Though both natural ecosystems and agricultural systems are sensitive to droughts, the agricultural systems are generally more delicate as the crops significantly vary in their water use efficiencies compared to natural vegetation [2]. Insufficient water availability in agricultural soils prevents supplementary nutrient absorption and impacts crop growth, gene expression, nutrient circulation, yield production and quality [3]. The surge in world population augments the dilemma projected to reach 9.8 billion people by 2050. Further, this population expansion is primarily expected in drought-prone areas of Africa and Asia [4]. It will pressure the global food market, suggesting cereal production must increase by 26%, which is 2.8 to 3.5 billion tons to current production. For now, improving the grain yield of agricultural crops seems a big challenge since drought spells are getting more frequent and scientific progress to combat it is relatively slow [5].

Seed priming is among reliable and efficient approaches for enhancing seed germination. The priming method involves treating seeds/plant organs with a variety of organic or inorganic compounds, visible and invisible electromagnetic radiations, and high or low temperatures [6]. Seeds undergo physiological changes during priming, i.e., regulation of hydration and aeration processes resulting in better and improved germination through metabolic progression [7]. Limited information is available on the application of physical priming approaches among the numerous priming techniques. However, there has been a rise in employing physical measures to stimulate plant growth in recent years [8]. In some reports, physical pre-planting treatments of seeds modulated change in the biochemical and physiological processes of seeds, consequently speeding up seedling growth and enhancing crop plant yield [9].

Photobiology is a branch of biology that studies the effects of lasers or other radiations on living organisms. Scientists are getting familiar with this technology and using it to augment their efforts in enhancing abiotic stress tolerance and crop yield. This technology has been more affordable for a wide range of farming communities and for large-scale use in agricultural fields [10]. Because of the essential properties of lasers, like polarization and intensity, laser radiation has a strong potential for use in photobiology. These features of laser radiation make it valuable in the biological and agricultural domains, where it can have multiple effects on crop plants [8,11]. Certain progressions in plant height, fresh and dry weights, and biochemical up-regulation, such as proteins and starch contents of plants, are triggered after laser radiation [12,13]. These effects can either prevent damaging effects of environmental adverseness through the improvement of the seed germination, depending upon the type of laser radiation used and duration of exposure time [14]. Drought stress prevents supplementary nutrient absorption and impacts crop growth, gene expression, circulation, and yield quality [3]. Plants respond to water stress by modulating various physiological processes, including photosynthesis, tricarboxylic acid cycle, sugar synthesis, glycolysis and hormone production [15].

Wheat is a major crop belonging to the family Poaceae in the order Triticale [16,17]. Even in high-income countries, wheat production is hindered by about 60% due to severe droughts. This condition is even worse in low-income countries where about 32% of the land, approximately 99 million hectares, is affected by severe droughts [18]. In both cases, a total crop failure may result from severe droughts and substantial yield losses. However, adaptive strategies such as organic osmolytes accumulation in root, stem and leaf tissues during the vegetative stage may contribute to stress tolerance in plants. Other strategies include maintaining a high root: stem ratio, high water use efficiencies, and redistributing the assimilates to the ear development, especially at the reproductive stage etc. [19].

Helium–neon (or He–Ne) and invisible infrared waveband ($CO_2$) lasers are the most commonly used radio agents to enhance crop stress tolerance. Previous studies have shown a positive role of lasers in accelerating crop metabolism, growth, and yield [20]. Laser pretreatment at suitable doses shows promising effects on seed germination capacity and enhances enzyme activities, pigments and photosynthetic efficiencies at the seedling stage. These effects boost vegetative growth and the seed yield at the reproductive stage in drought stressed crops [21]. For example, Qiu et al. [22] characterized miRNAs and regulation of their target genes by He–Ne laser priming of wheat seedlings under drought stress. They found that the plants originated from He–Ne laser primed seeds have significantly higher levels of antioxidative enzymes, relative tissue water and malondialdehyde contents than their counterpart control plants. This superior performance of laser-primed plants was significantly linked to their capacity to down-regulate several miRNA species. In another study, Qiu et al. [23] showed that laser-priming significantly promoted drought tolerance in wheat seedlings by upregulating the key transcriptional genes linked with superior photosynthetic capacity, nutrient acquisition and transport, and reactive oxygen species (ROS) homeostasis. Similarly, Ali et al. [24] suggested that the enhanced drought tolerance potential of *Celosia argentea* was linked to changes in fatty acids, phenolics, and antioxidant profiles regulated by pre-sowing He–Ne laser seed irradiation.

Given the stimulatory roles of radio-priming on different plant species, we hypothesized that laser (He–Ne) pre-sowing treatment would protect wheat seedling growth and physiological process from drought injury. The questions addressed in this work were: To what degree does laser (He–Ne) pre-sowing treatment induce water stress tolerance in wheat, and is it equally effective in wheat as reported for other crops? The findings of this study will help understand the biochemical and physiological mechanisms regulated by laser exposure and to conclude the commercial applicability of lasers to enhance wheat productivity in drought prone areas.

## 2. Materials and Methods

### 2.1. Seed Material and Laser Irradiation Treatment

The effect of laser seed priming (LSP) on wheat under water stress was evaluated. Two varieties of wheat (V1 = FSD-2008 and V2 = Anaj-2017) were obtained from the Ayub Agriculture Research Institute (AARI), Faisalabad and sown in plastic pots during the winter season of 2020–2021. Prior to sowing, the seeds were soaked in $dH_2O$ in beakers for 3 h. The seeds were then sterilized with 0.01% $HgCl_2$ and air-dried. The seeds were pretreated with laser light at 630 nm for 1, 2 and 5 min, while control seeds were not treated with laser. The pots were filled with 10 kg of sandy loam textured soil and arranged in a completely randomized design in three replicates. Initially, ten seeds were sown in each pot, and three plants per pot were maintained after germination. After 7 days of germination, a 50% field capacity level was maintained along with control irrigation treatment, i.e., 100% field capacity level. The water stress was maintained by daily weighing the pots, and the required amount of water was added to each pot. The pots were placed in the Botanical Garden, the University of Agriculture, Faisalabad, under the following conditions: 10/14 h day length; $14 \pm 3$ °C mean temperature and $49.7 \pm 7$% mean relative humidity. Water-stressed pots were covered with plastic sheets to avoid accidental moisture from rain or dew. The pots were supplied with a recommended dose of N, P and K fertilizers.

### 2.2. Growth Parameters

After 21 days of water stress, the plants were harvested to measure growth parameters such as shoot and root length (cm), fresh weight of shoot and root (g), dry weight of shoot and root (g), the number of leaves per plant and area of leaf per plant ($cm^2$). The plants were carefully harvested, and soil in the rooting zone was removed by running water. The roots and shoots were detached, and their lengths were measured with a scale. A digital balance was used to take fresh weights, and dry weights were measured after complete drying in an oven at 80 °C for one week. The number of leaves were counted. The leaf area (LA) was measured by using a formula devised by Lopes et al. [25]:

$$Area = length \times width \times correction\ factor\ (0.75) \tag{1}$$

The selected plants were grown to maturity under the same experimental conditions outlined above, and their yield was measured. The total number of tillers and fertile tillers were counted at crop maturity. The grains were harvested and used to determine 100-grain weight.

### 2.3. Photosynthetic Pigments

Chlorophyll '*a*', and '*b*' were measured by the Arnon's method [26] while carotenoids were measured by Davis's method [27]. A 0.1 g fresh leaf material was extracted in 10 mL of 80% acetone, and measurements were read at 663 and 645 nm using a spectrophotometer (Hitachi-U 2001, Tokyo, Japan). The concentration of photosynthetic pigments was estimated using the equations provided by Arnon [26] and Davis [27].

### 2.4. Anthocyanin

Anthocyanin was measured according to Krizek et al. [28]. A 0.2 g of leaves sample was homogenized with 3 mL of 1% HCL–methanol (99:1). The extract was centrifuged

at 18,000× $g$ for 30 min at 3 °C. The supernatant was left overnight in the dark at 5 °C. Anthocyanin content was measured at 550 nm using a spectrophotometer (Hitachi-U 2001, Tokyo, Japan). The extinction coefficient for anthocyanin was 33,000 cm$^{-2}$ mol$^{-1}$. The final concentration of anthocyanins measured was expressed as mg/g fresh weight.

## 2.5. Enzymatic Antioxidants

An extract of the plant was prepared to assess the antioxidant enzyme. For extraction, a 0.5 g fresh plant leaf material was pulverized in 10 mL of 50 mM cooled potassium phosphate buffer (pH 7.8). This homogenized mixture was centrifuged for 20 min at −4 °C at 15,000× $g$. The debris was removed, and the supernatant was used to test the activity of various enzymatic antioxidants.

Catalase (CAT) activity was measured using the protocol as described by Chance and Maehly [29] with some alterations. A 1.9 mL potassium phosphate buffer with pH 7.8 was mixed with 1 mL H$_2$O$_2$ to make CAT reaction mixture. The reaction was started by the addition of 0.1 mL enzyme extract to the above mentioned mixture. After 20 s, the reduction in absorbance at 240 nm was read with a spectrophotometer (Hitachi-U 2001, Tokyo, Japan). An alteration in absorbance of 0.01 units per minute was defined as 1 unit of CAT activity.

With some modifications, peroxidase (POD) activities were detected by the procedure described by Chance and Maehly [29]. A 100 μL H$_2$O$_2$, 750 μL potassium phosphate buffer having pH 7.8, 50 μL of the sample, and 100 μL guaiacol were used to start the reaction. After that, the change in absorbance at 470 nm was measured at 20 s with a spectrophotometer (Hitachi-U 2001, Tokyo, Japan). A 0.01 absorbance change per minute per mg of protein was used to define one unit of POD activity.

The Spitz and Oberley [30] method was used to determine the superoxide dismutase activity (SOD). For this purpose, a 2 mL plastic cuvette was taken, and 400 μL H$_2$O + 250 μL potassium phosphate buffer (pH 7.8), 100μL methionine, 100 μL triton, 50 μL nitro blue tetrazolium 50 μL enzyme extract were mixed. Then 50 μL riboflavin was added and placed under the fluorescent lamp for about 15 min to induce the superoxide dismutase reaction. After that, SOD activity was measured by a spectrophotometer (Hitachi-U 2001, Tokyo, Japan) at 560 nm, and it was expressed in mg$^{-1}$ protein.

For the measurement of ascorbate peroxidase (APX), the extract of enzyme corresponding to 150 μg protein from plant leaf was added to a reaction mixture containing 100 mM potassium phosphate buffer (pH 7.8), 0.2 mM H$_2$O$_2$ and 0.5 mM ascorbic acid. The reaction was allowed to proceed for 1 min, and the reduction in absorbance at 290 nm was recorded with a spectrophotometer (Hitachi-U 2001, Tokyo, Japan). The APX activity was expressed in units of mg$^{-1}$ protein [31].

## 2.6. Soluble Sugars

Soluble sugar was measured using the Dubois et al. method [32]. A 0.1 g fresh plant material was boiled at 90 °C with 5 mL distilled water. After that, 9 mL of dH$_2$O was added to one mL of extract. A 0.5 mL sample of this solution was taken in 5 mL of anthrone reagent. The mixture was boiled in a water bath for 20 min at 90 °C, and then a spectrophotometer (Hitachi-U 2001, Tokyo, Japan) was used to take the absorbance at 620 nm. The final concentration of soluble sugars was determined using a standard curve developed with known glucose standards.

## 2.7. Soluble Proteins

The soluble proteins of the sample were determined using Bradford's method [33]. An amount of 0.5 g of fresh plant leaf tissues were crushed in 5 mM ice-cold phosphate buffer having pH 7.8 to extract protein. The homogenate was centrifuged at −4 °C for 15 min at 15,000× $g$. The supernatant was separated and used for protein extraction. A test tube was filled with 0.1 mL sample extract and 5 mL Bradford regent, carefully vortexed, and set aside for 30 min. These sample solutions and the blank were heated at 37 °C for

10–15 min, and absorbance was measured at 595 nm on a spectrophotometer (Hitachi-U 2001, Tokyo, Japan). The final concentration of proteins was determined using a standard curve developed from the known bovine serum albumin (BSA) standard.

### 2.8. Total Free Amino Acids

Total free amino acids were determined according to Hamilton and Van Slyke [34]. To evaluate total free amino acids (TFA) levels, 2 g ninhydrin was dissolved in 100 mL $dH_2O$, and a 10% pyridine solution was prepared. A total of 1 mL of extract was placed in a test tube and 1 mL of 10% pyridine added, and then 1 mL of 2% ninhydrin solution in each test tube. The test tubes were heated in a boiling water bath for half an hour and the volume increased to 50 mL with $dH_2O$. The measurements were taken at 570 nm using a spectrophotometer (Hitachi-U 2001, Tokyo, Japan). The final TFA concentration was determined using a standard curve developed from the known leucine standard.

### 2.9. Ionic Contents of the Plant Sample

The 0.1 g dried plant material was acid—digested ($H_2SO_4$:$H_2O_2$ method) using the method of Wolf [35]. A flame photometer (Jenway PFP-7, England) was used to determine the amounts of K, Na and Ca in the digested sample. The standard curves were developed using a graded series of Na, K and Ca standards ranging from 10 to 100 ppm. The actual quantities of the mineral elements were computed and expressed as mg $g^{-1}$ of dry plant material after comparing them to the standard curves.

### 2.10. Statistical Analysis

The experimental design was a complete randomized design (CRD) with three replicates. The main effects of laser treatment (factor 1), irrigation regime (factor 2) and varieties (factor 3) and their interaction were determined by three-way ANOVA using Costat computer package. The least significant differences (LSD) values were computed and used to compare the significance of means at $p \leq 0.05$ according to the method of Snedecor and Cochran [36]. Heatmaps and correlograms were constructed using customized codes in R Studios (R Version 1.1.463).

## 3. Results

### 3.1. Growth and Yield Attributes

Moisture deficit caused a significant reduction in root and shoot length, leaf area, and leaves per plant (Figure 1). The root and shoot lengths were substantially higher in laser primed plants (up to 2 min exposure time) grown under moisture deficit conditions (50% FC level). The 5 min exposure time did not cause any significant enhancement in root length and leaves per plant but shoot length and leaf area were substantially higher in moisture deficit conditions than in unprimed control plants. In water-stressed conditions, a 2 min exposure time caused a maximum increase in these attributes of cultivar Anaj-2017, which differed significantly from FSD-2008 at the same treatment level. The leaf area in cultivar FSD-2008 under 5 min laser exposure time was even lower than the unprimed control, while Anaj-2017 had substantially higher leaf area at the same priming level.

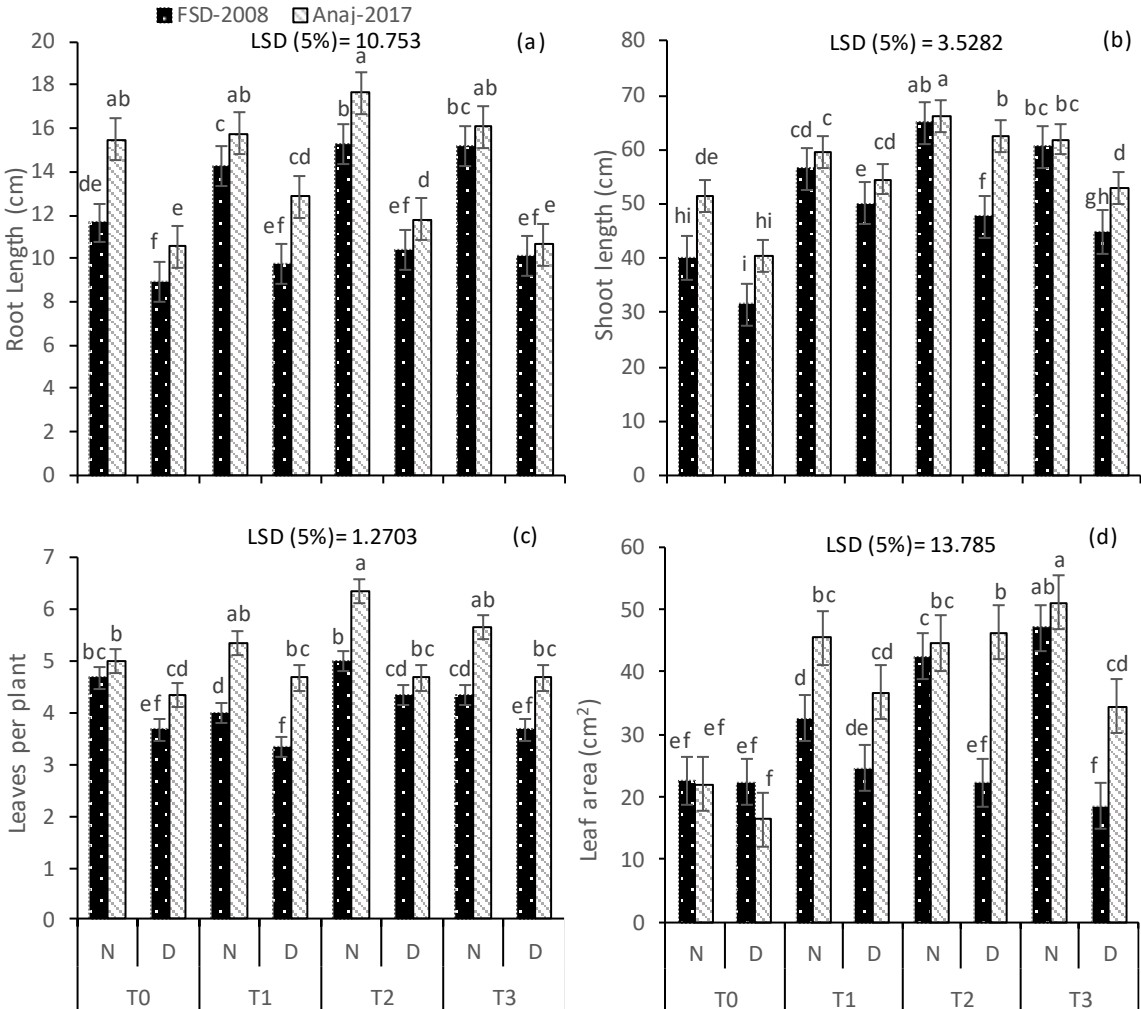

**Figure 1.** Effects of laser seed priming on (**a**) root length, (**b**) shoot length, (**c**) number of leaves per plant and (**d**) leaf area in wheat under drought stress. Legend: N = Control (100% field capacity), D = Drought (50% field capacity), T0 = Unprimed control, T1 = 1 min laser priming, T2 = 2 min laser priming and T3 = 5 min laser priming. Means sharing same lowercase letters on bars are statistically non-significant at $p \leq 0.05\%$).

All laser priming levels (1 to 5 min) caused a significant enhancement in root and shoot fresh and dry weights of both wheat genotypes under 50% FC level (Figure 2). The maximum root fresh and dry weights were recorded in 1 and 2 min laser-treated Anaj-2017 plants grown under normal conditions, while in FSD-2008, it was at the maximum only in the 2 min exposure time. Under water stress, both genotypes showed the maximum but statistically non-significant differences in the fresh root weight for 2 and 5 min of laser exposure time. Shoot fresh and dry weights were the maximum in both normal (100% FC), and water-stressed (50% FC) plants under 2 min exposure time. Though both these attributes decreased in the water stress under 5 min exposure time, they were substantially higher than non-laser treated plants subjected to water stress conditions. Overall, the improvement in these growth attributes was significantly higher in Anaj-2017 than in FSD-2008.

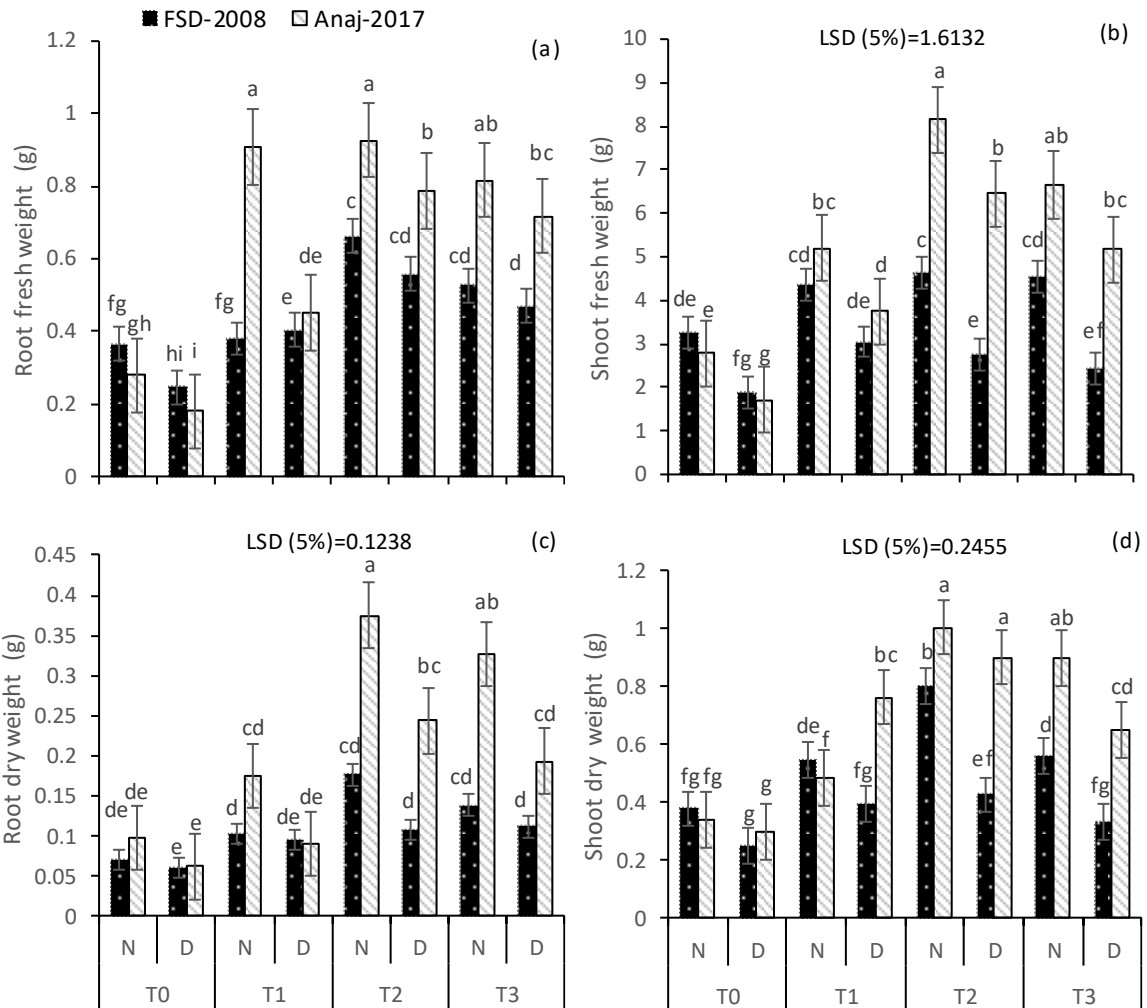

**Figure 2.** Effects of laser seed priming on (**a**) root fresh weight, (**b**) shoot fresh weight, (**c**) root dry weight and (**d**) shoot dry weight in wheat under drought stress. Legend: N = Control (100% field capacity), D = Drought (50% field capacity), T0 = Unprimed control, T1 = 1 min laser priming, T2 = 2 min laser priming and T3 = 5 min laser priming. Means sharing same lowercase letters on bars are statistically non-significant at $p \leq 0.05\%$).

The yield attributes were substantially higher in Anaj-2017 than in FSD-2008 under control and water-stressed conditions (Figure 3). Laser priming significantly improved yield attributes of the two wheat genotypes under both moisture deficit conditions.

Under moisture deficit conditions (50% FC), the highest total number of tillers and fertile tillers per plant were produced in the 2 min laser exposure time, resulting in significantly higher yield (100 grain weight) at the same treatment levels. One min exposure time has a non-significant effect on these growth attributes compared with un—primed controls. Though the longest exposure time (5 min) caused some enhancement in yield attributes, it was significantly lower than 1 or 2 min exposure times and almost equivalent to unprimed, non-stressed control plants.

In FSD-2008, shoot (33.84%) and root (26.52%) lengths, plant height (27.54%), root (38.67), and shoot dry (46.77%) weights were maximally improved under 2 min laser priming time under water stress. In Anaj-2017, root fresh (50.18%) and dry weight (48.57%) and shoot dry weight (57.93%) were the maximum water stress under 1 min laser priming. The root length and flag leaf area in water stress were increased by 33.64% and 32.49%, respectively, under 5 min laser priming. In both FSD-2008 and Anaj-2017, the total number of tillers and fertile tillers increased by 50% under 1 min laser priming treatment. The 100 g grain weight was more in FSD-2008 (34.01%) than in Anaj-2017 (28.14%).

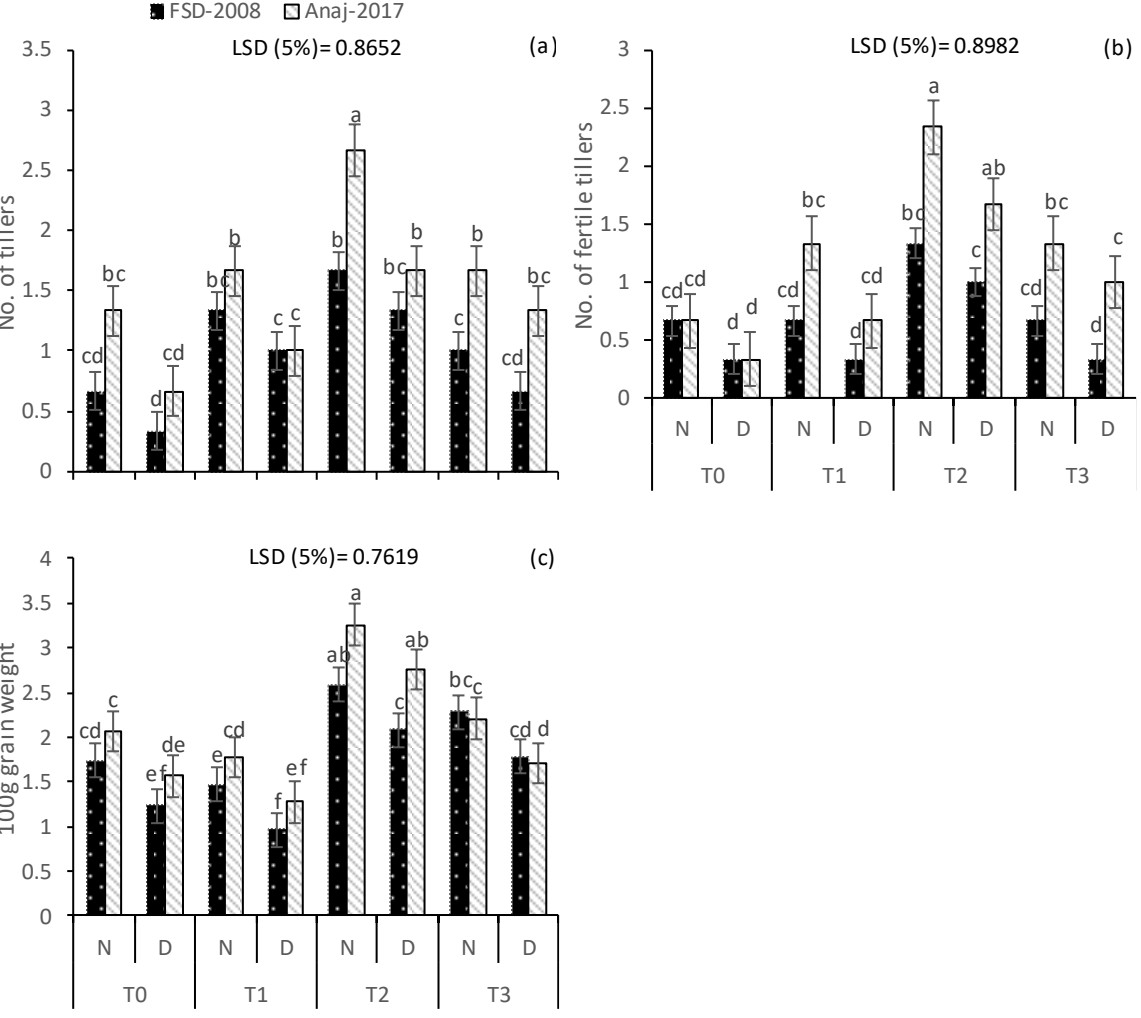

**Figure 3.** Effects of laser seed priming on (**a**) number of tillers, (**b**) number of fertile tillers and (**c**) 100 g grain weight in wheat under drought stress. Legend: N = Control (100% field capacity), D = Drought (50% field capacity), T0 = Unprimed control, T1 = 1 min laser priming, T2 = 2 min laser priming and T3 = 5 min laser priming. Means sharing same lowercase letters on bars are statistically non-significant at $p \leq 0.05\%$).

### 3.2. Photosynthetic Pigments

Chlorophyll *a* content was a little affected in all laser treatment levels under non-stressed conditions (Figure 4). However, a significant decrease was observed in water-stressed plants, which was almost equally mitigated by 2 and 5 min exposure times. The 2 min laser priming effectively enhanced chlorophyll *b*, total chlorophyll and anthocyanin, whereas 5 min exposure time was relatively less effective in improving these photosynthetic attributes. The chlorophyll *a/b* ratio in both varieties was higher in unprimed plants in both normal and water-stressed conditions. It decreased under 2 min laser priming but was substantially higher in 5 min exposure time. Carotenoids increased gradually under 2 min laser priming but remained constant under 5 min exposure time. In cultivar FSD-2008, chlorophyll *a* (12.69%), carotenoids (39.97%) and anthocyanin (37.83%) achieved the maximum value in water stress plants with 1 min laser priming. While under 5 min laser priming time, the chlorophyll *b* and chlorophyll *a/b* increased by 37.84% and 38.52% under water stress, respectively. In Anaj-2017, chlorophyll *b* and carotenoid contents were maximum under 2 min laser priming, i.e., 50.68% and 13.01%, respectively, in water stress, compared with their respective controls. Under 5 min laser priming under water stress, chlorophyll *a/b* and total chlorophyll contents were 33.72% and 30.47% higher, respectively.

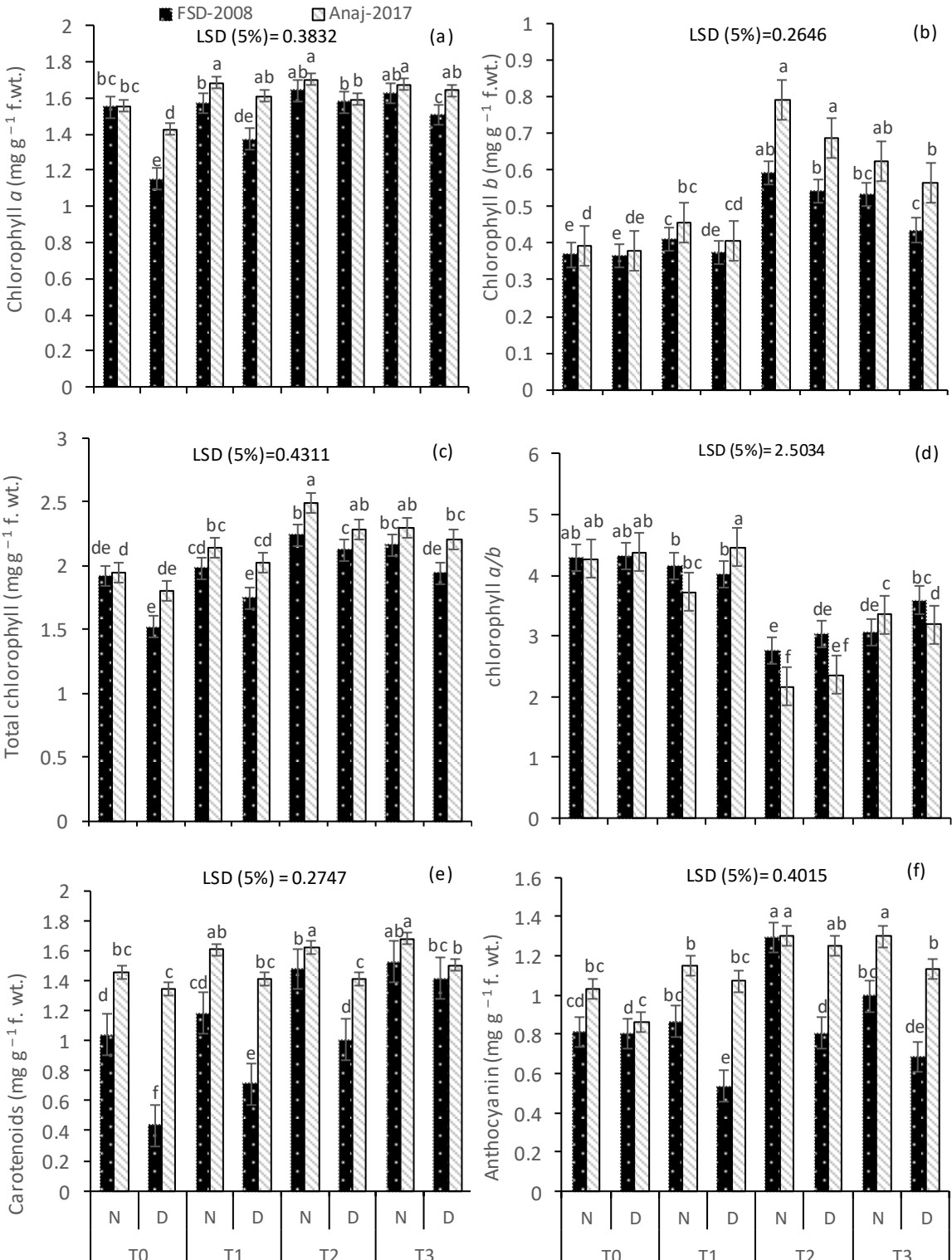

**Figure 4.** Effects of laser seed priming on (**a**) chlorophyll *a*, (**b**) chlorophyll *b*, (**c**) total chlorophyll, (**d**) chlorophyll *a/b*, (**e**) carotenoids and (**f**) anthocyanin in wheat under drought stress. Legend: N = Control (100% field capacity), D = Drought (50% field capacity), T0 = Unprimed control, T1 = 1 min laser priming, T2 = 2 min laser priming and T3 = 5 min laser priming. Means sharing same lowercase letters on bars are statistically non-significant at $p \leq 0.05\%$).

### 3.3. Antioxidative Enzymes

The activities of all antioxidative enzymes (SOD, POD, CAT and APX) were substantially higher in both genotypes growing under 50% field capacity at all laser-priming levels. Anaj-2017 exhibited substantially higher activities of these antioxidative enzymes compared with cultivar FSD-2008. Overall, the 2 min laser pretreatment time was more effective in activating the enzymatic antioxidative defense system under 50% field capacity level. The 2 and 5 min laser exposure times showed almost equal enhancement in these enzymatic antioxidants except for APX, which was substantially higher in 5 min exposure time. In water-stressed cv. FSD-2008, SOD and CAT experienced a minimum increase under 2 min laser priming (56.12% and 71.38%, respectively), while POD activity was minimally increased (41.48%) under 5 min laser priming. The APX was minimally (11.31%) alleviated under 1 min laser priming under water stress. In Anaj-2017, SOD, POD and CAT activities were maximum under 5 min laser priming, i.e., 45.81%, 1.34% and 38.25%, respectively. The APX was increased by 3.53% in this genotype when water stress was applied at 50% field capacity, and seeds were exposed to laser priming for 1 min intervals (Figure 5).

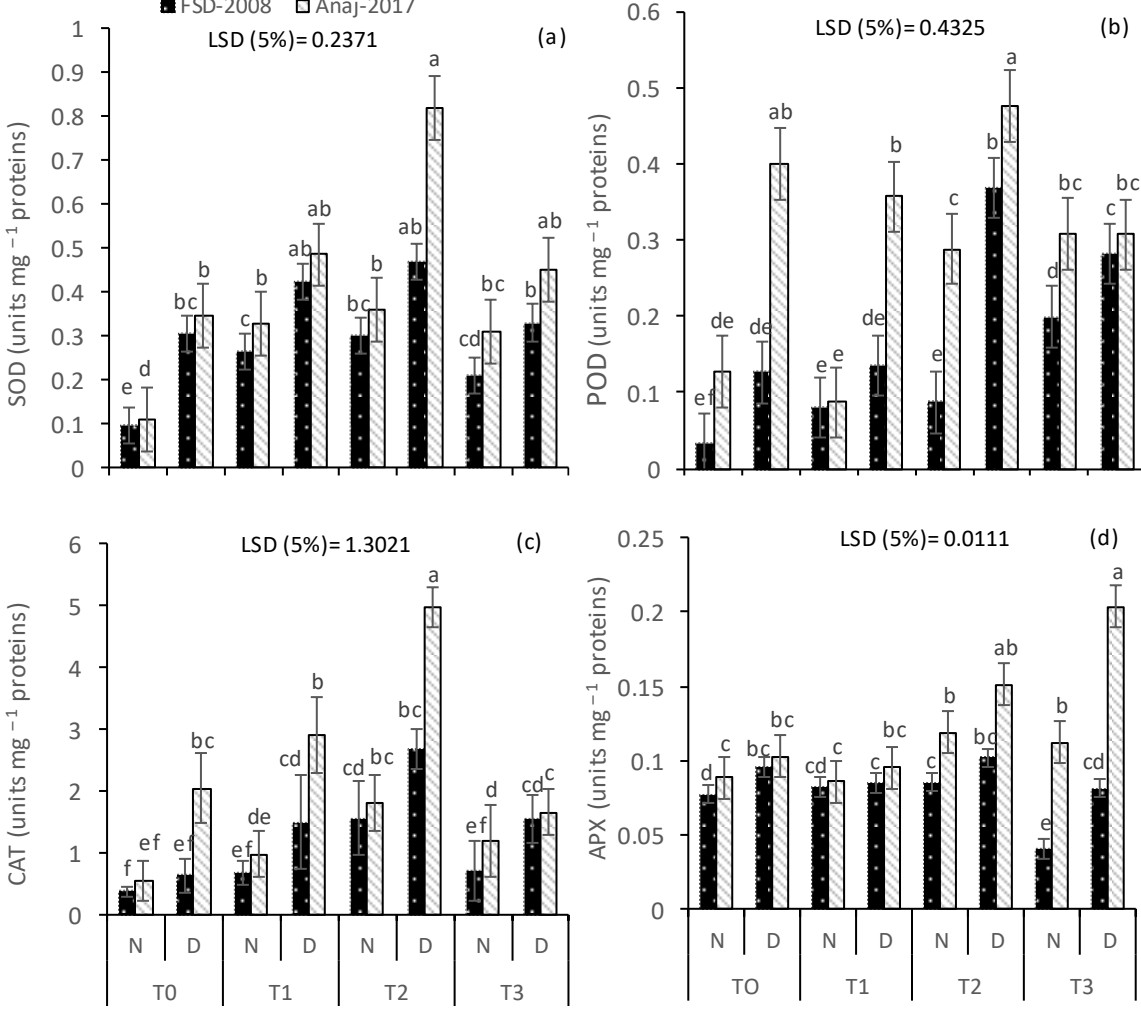

**Figure 5.** Effects of laser seed priming on (**a**) SOD, (**b**) POD, (**c**) CAT and (**d**) APX in wheat under drought stress. Legend: N = Control (100% field capacity), D = Drought (50% field capacity), T0 = Unprimed control, T1 = 1 min laser priming, T2 = 2 min laser priming and T3 = 5 min laser priming. Means sharing same lowercase letters on bars are statistically non-significant at $p \leq 0.05\%$).

### 3.4. Soluble Proteins and Free Amino Acids

Total soluble proteins and free amino acids were enhanced to a lesser extent by laser treatment of seeds grown, with a slightly better result obtained under 2 min exposure time under control or 50% field capacity levels. Total soluble sugars were slightly higher under 1 min exposure time. In FSD-2008, under 2 and 1 min laser priming caused a maximum (17.23%) in total soluble proteins and total soluble sugars (8.72%), respectively, compared with control. Further, 2 min laser priming caused a maximum increase in total free amino acids (14.66%) under water stress environments. In Anaj-2017, under 1 min laser priming time, the total soluble sugars and free amino acids were maximally raised under water stress, i.e., by 40.01% and 22.91%, respectively (Figure 6).

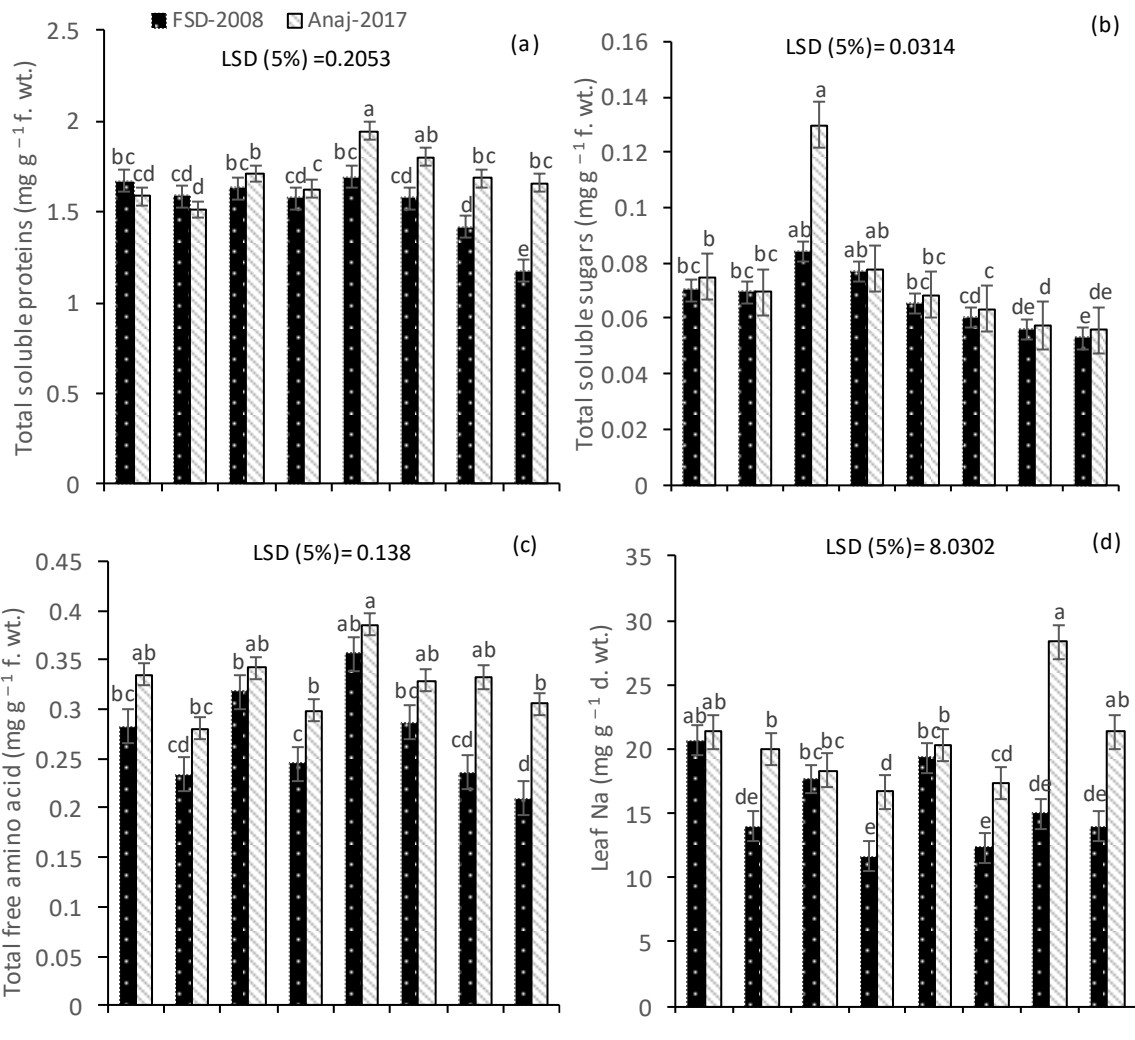

**Figure 6.** *Cont.*

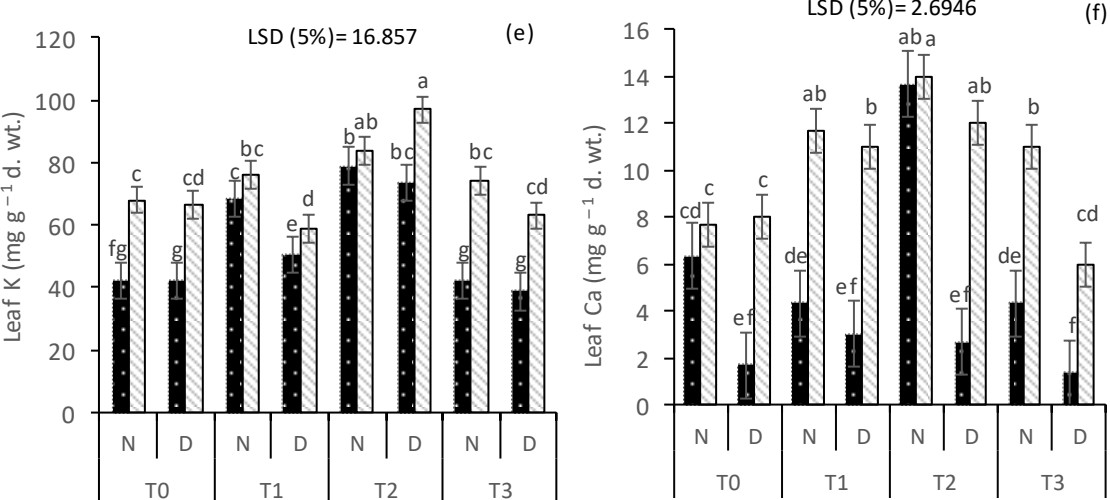

**Figure 6.** Effects of laser seed priming on (**a**) total soluble proteins, (**b**) total soluble sugars, (**c**) total free amino acids, (**d**) leaf sodium ion, (**e**) leaf potassium ion and (**f**) leaf calcium ion in wheat under drought stress. Legend: N = Control (100% field capacity), D = Drought (50% field capacity), T0 = Unprimed control, T1 = 1 min laser priming, T2 = 2 min laser priming and T3 = 5 min laser priming. Means sharing same lowercase letters on bars are statistically non-significant at $p \leq 0.05\%$).

### 3.5. Leaf Ionic Contents

Leaf Na contents were maximally enhanced under 5 min followed by 2 min exposure times. Leaf K and Ca were the maximum in 2 min laser pre-sowing treatment (Figure 6). Leaf sodium was maximally alleviated (32.25%) in 0 min laser priming at 100% field.

### 3.6. Heatmap Clustering

Heatmap cluster analysis was done to represent a two-way interaction between different laser seed priming and water stress levels. In the heatmap, the color of the individual cells displayed as boxes shows the interaction strength between attributes and drought treatments. The strength of the color scale from black (most negative) to red (most positive) was directly proportional to the strength of color gradient used in the heatmap. Hierarchical clustering was used to group parameters (in columns) against treatment (in rows) under the influence of various laser treatments to water stress seedlings.

Clustered heatmap among morphological, physiological, biochemical and yield parameters constructed for wheat cultivar FSD-2008 generated four distinct clusters. The 2 and 3 min. laser priming levels were grouped in two cluster at control (100%) and 50% field capacity levels. Here most of the recorded attributes were strong positively correlated at 100% FC level while weekly correlated with 50% level. Similarly, unprimed and 1 min. laser treatments were closely grouped in control and 50% field capacity levels. For plant attributes, the largest cluster included the number of tillers closely grouped with plant height, shoot length, root fresh weight and root dry weight. The number of fertile tillers and 100 g grain weight were closely related with anthocyanin, leaf number, leaf calcium, total chlorophyll and plant height. In cluster 2, flag leaf area, shoot fresh weight, shoot dry weight and root length were closely grouped with chlorophyll *a* and *b*, chlorophyll *a/b*, carotenoids, total free amino acids, and leaf sodium. In cluster 3 leaf potassium was closely grouped with total soluble sugar and total soluble proteins, The cluster 4 containing all anti-oxidative enzymes like superoxide dismutase, peroxidase, catalase and ascorbate peroxidase was a separate cluster branching above all other clusters that indicated their collective role in modulating growth, physiological and yield attributes (Figure 7a).

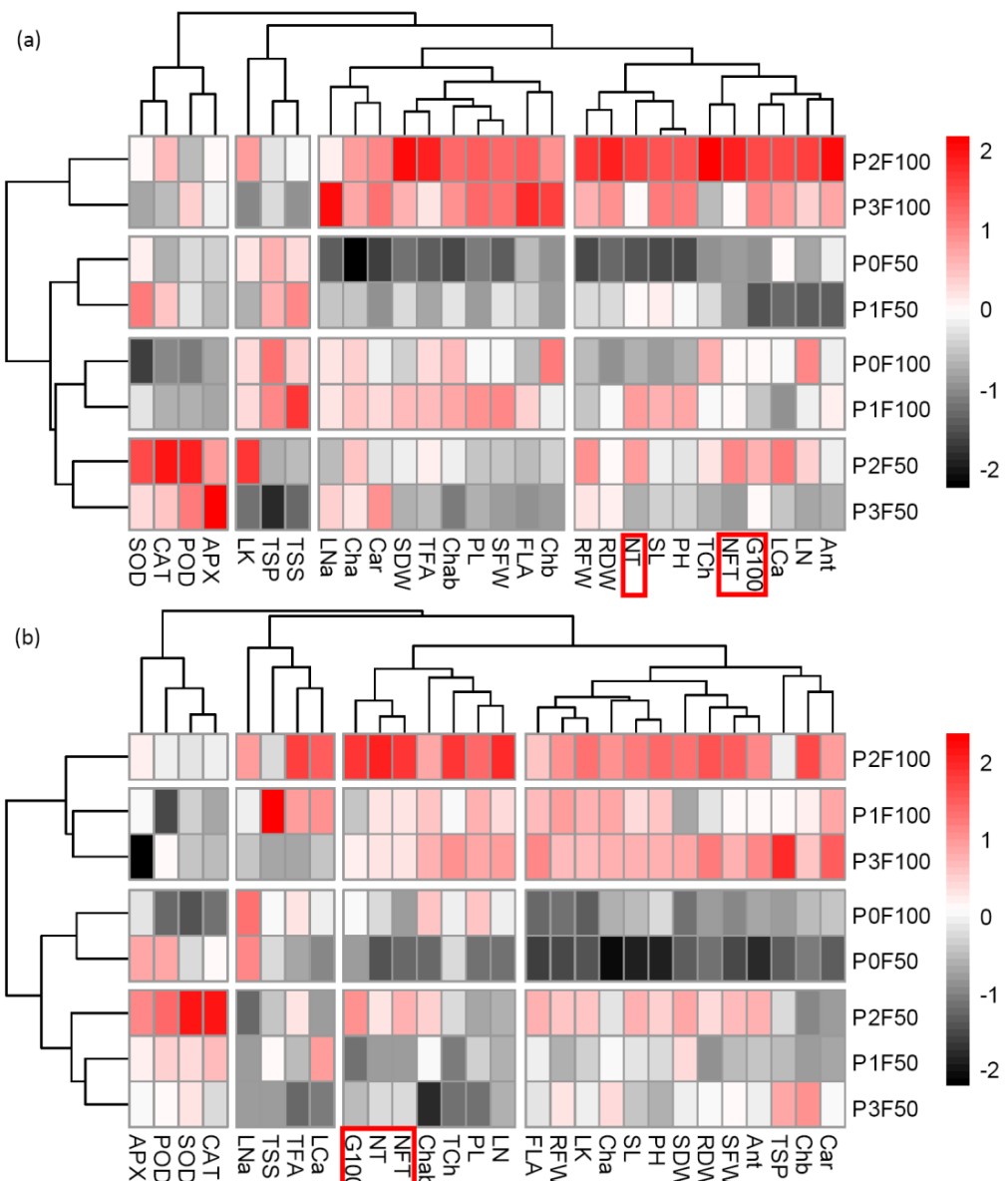

**Figure 7.** Heatmap clustering shows the growth and yield regulation of (**a**) FSD-2008 and (**b**) Anaj-2017 wheat varieties through laser priming-induced modulation of physiological attributes under drought stress. The yield attributes are included in the box(es). Priming: P0F100 = unprimed seeds with 100% field capacity (FC), P0F50 unprimed seeds with 50% FC, P1F100 = 1 min priming with 100% FC, P1F50 = 1 min priming with 50% FC, P2F100 = 2 min priming with 100% FC, P2F50 = 2 min priming with 50% FC, P3F100 = 5 min priming with 100% FC, P3F5 0 = 1 min priming with 50% FC. Plant attributes: PL = plant root length, SL = plant shoot length, PH = plant height, RFW = root fresh weight, SFW = shoot fresh weight, RDW = root dry weight, SDW = shoot dry weight, LN = no. of leaves, FLA = flag leaf area, Cha = chlorophyll *a*, Chb = chlorophyll *b*, Chab = chlorophyll *a/b*, TCh = total chlorophyll, Car = carotenoids, Ant = anthocyanin, TSP = total soluble proteins, TSS = total soluble sugars, TFA = total free amino acids, SOD = superoxide dismutase, POD = peroxidase, CAT = catalase, APX = ascorbic acid peroxidase, LNa = leaf sodium ions, LK = leaf potassium ions, LCa = leaf calcium ions, NT = no. of tillers, NFT = no. of fertile tillers and G100 = 100 g grain weight. capacity level against 50% filed capacity level in FSD-2008. Leaf potassium was increased by 26.69% under 2 min laser priming in water-stressed plants, while leaf calcium was maximally increased (45.45%) at the same (2 min) laser exposure time. In Anaj-2017, leaf Ca and Na contents were reduced by 36.20% and 80.95% under 2 min laser priming level under drought stress (50% field capacity).

For wheat variety Anaj-2017, 1, 2 and 5 min laser priming levels at 50% field capacity level clustered in a separate cluster while 1 and 5 min laser priming levels under 100% field capacity level grouped in another cluster. The unprimed plants were clustered separately at both field capacity levels. Plant growth attributes, such as flag leaf area, plant height, shoot length, root fresh and dry weight, shoot fresh, dry weight and leaf potassium ion were clustered with photosynthetic pigments chlorophyll *a* and *b*, carotenoids, and with anthocyanin and total soluble proteins. Cluster 2 contained yield attributes like the number of tillers, numbers of fertile tillers, 100 g grain weight closely grouped with the number of leaves, root length, and chlorophyll *a/b* ratio. Cluster 3 showed a close grouping of total free amino acids and total soluble sugars with leaf calcium and leaf sodium. Cluster 4, containing antioxidative enzymes (SOD POD, CAT and APX), was branched over all other three clusters indicating their critical roles in modulating the growth and physiological attributes (Figure 7b).

### 3.7. Pearson Correlation Coefficient

In FSD-2008, most growth attributes were strongly positively correlated with physiological attributes and concentration of photosynthetic pigments. In contrast, the activities of antioxidative enzymes were not correlated with any growth, physiological or yield attribute. Leaf nutrients were positively correlated. The fertile tiller numbers were strongly positively linked to the leaf number and shoot dry weight, which in turn, were positively correlated with the total number of tillers per plant. The 100 g grain weight was positively correlated with root and shoot dry weight, number of leaves, carotenoids, anthocyanins and total chlorophyll (Figure 8a). Likewise, in Anaj-2017, all growth and physiological attributes were strongly positively correlated with the number of tillers produced per plant. Activities of antioxidative enzymes were strongly negatively correlated. The 100 g grain weight was positively correlated with shoot length, shoot fresh and dry weights that were correlated with the number of fertile tillers (Figure 8b).

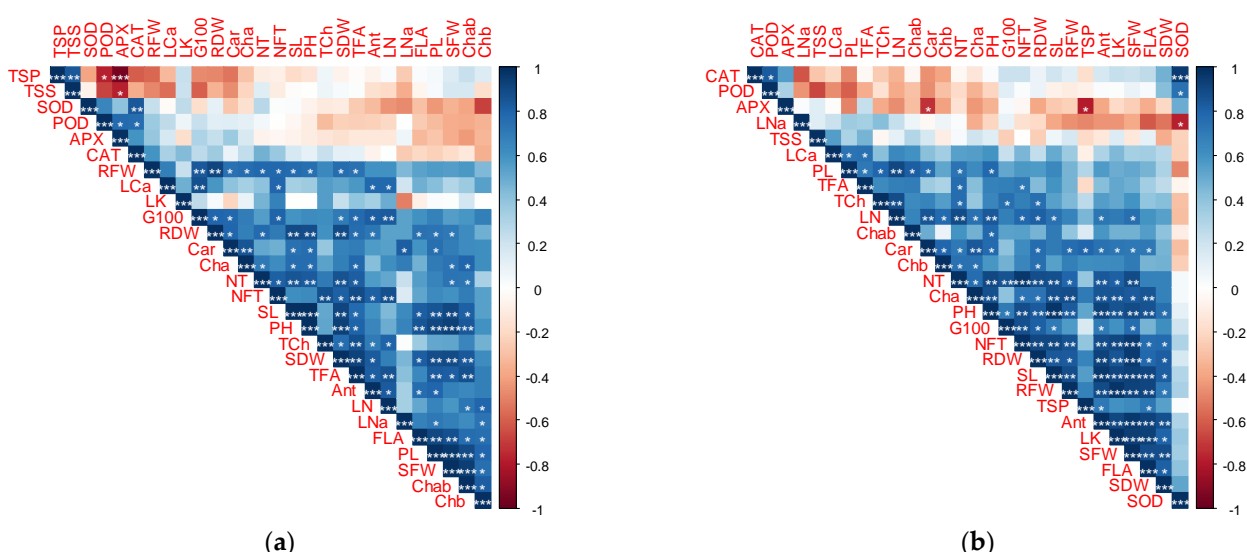

**Figure 8.** Pearson's correlation matrixes constructed for growth, physiological and yield attributes of laser-primed (**a**) FSD-2008 and (**b**) Anaj-2017 wheat varieties under drought stress. For abbreviations of priming levels and plant attributes, please see the caption of Figure 7. * significant at $p \leq 0.05\%$; ** significant at $p \leq 0.01\%$; *** significant at $p \leq 0.001\%$.

## 4. Discussion

A significant effect of water stress on growth attributes was observed where seedling shoot-root lengths and weights decreased. Laser seed pretreatment substantially enhanced the plant water-stress tolerance and final yield (especially the number of fertile tillers and 100 grain weight). The maximum grain yield increase was achieved under 2 min

laser priming in this study. Previously, Qiu et al. [23] showed that an increase in the time duration of laser priming affected the growth parameters. An increase in irradiance intensity and exposure time of He–Ne laser has been reported to increase the levels of endogenous growth-promoting substances (like indole acetic acid and gibberellic acid) with a concurrent decrease in growth-inhibiting (phenol and abscisic acid) substances in germinating seeds [37]. As observed in this study, increased endogenous levels of growth-promoting substances positively affected seedling growth and induced water stress tolerance in wheat plants. Similarly, Metwally et al. [38] reported that He–Ne laser priming could significantly enhance yield and fatty-acid contents of *Caster bean* under drought stress.

A significant decrease in the total number of leaves per plant was seen in the current experiment in water stress that increased under 2 min laser seed-priming. Phytochrome, a photoreceptor, shows absorption spectra in the far-red and red light of the visible spectrum. This receptor has well-known regulatory effects on many plant responses like seed germination, seedling elongation, shape, size, and leaf number [21]. Laser priming also influences phytochromes and induces almost parallel growth responses [39]. Our study also showed that there was a significant decrease in leaf area under water stress that was maximally increased by 5 min laser priming. In a past study, Rybinski and Garczyński [40] observed that maize, barley, and wheat plants originating from laser-irradiated seeds had a substantially larger leaf surface area compared with those grown from non-irradiated seeds. Such a large leaf area during the vegetative phase of plant growth grown from the laser-irradiated seeds was due to faster growth rates than the non-laser-treated plants. The findings of our study are in agreement with Ali et al. on *Celosia argentea* [41], Al-Sherbini et al. on pea [42] and AlSalhi et al. on wheat [43].

Modern studies show that laser priming can enhance chlorophyll contents. Qiu et al. [23] also found similar results where laser pretreatment upregulated most genes related to photosynthesis, including proteins (PsbR) of PS−II, ATP synthase and chlorophyll-binding protein. The laser exposure may also improve ATPase activity through increasing energy. A laser photon has 1.96 eV of energy, approximately equal to sixty high-energy phosphate bonds [44]. Anthocyanins were also increased in the current study, as already confirmed by Abou-Dahabe et al. [45]. These findings suggest that chlorophyll *a*, chlorophyll *b* and total chlorophyll predominantly increased until 2 min laser priming, but decreased under 5 min exposure. Carotenoids increased to the maximum value under 5 min laser priming. Pigments always give a good response when exposed to the lights. While the laser is also a monochromatic light, it induces better pigmentation, increasing photosynthetic efficiency. Thus, laser pretreatment results in progressive growth enhancement through increased photosynthates necessary for plant growth and final yield. Rosema et al. [46] showed a significant correlation between exposure to laser and chlorophyll fluorescence and enhancement in photosynthetic attributes of *Populus nigra*.

The antioxidant enzyme activities such as CAT, POD, SOD and APX were significantly increased in water-stressed plants. Laser priming under 2 min exposure substantially enhanced the activities of these antioxidative, although APX activity was elevated up to 5 min laser priming [47]. Laser radiation activated APX, POD, SOD, and CAT, and elevated total soluble proteins. This upregulated antioxidant system scavenged the uncontrolled ROS, thereby protecting photosynthetic pigments, metabolic functioning, macromolecules and cellular organelles from drought-induced oxidative damage. The maintenance of these processes was the key to enhancing growth parameters directly linked to yield [47]. The current study showed a parallel increase in protein, sugar and free amino acids. After 1 min, there was a consistent decrease in these parameters. In some past studies, a low dose of laser seed pretreatment had stimulatory effects on plant height, root length, and soluble sugar and soluble protein levels during the process of selecting efficient doses of He–Ne laser treatment for crops [48]. In this context, Zhu et al. [43] showed that He–Ne laser priming irradiation can effectively protect wheat plants from cadmium toxicity by modulating nutrient acquisition and antioxidative defense systems.

In the present study, with increasing laser exposure time, there was an increase in $Na^+$, $K^+$ and $Ca^{2+}$ ions in plant tissues. Similar results were obtained earlier in sugar beet [49], suggesting that laser priming could promote ion-binding genes [50]. Podleśny [51] reported that faba bean plants grown from laser-irradiated seeds produced heavier seeds, probably because increased pod length significantly changed the number of seeds per pod. Adverse effects of drought on wheat yield potential may be due to water stress effects by decreasing absorption of essential ions and water through a decrease in osmotic potential in soil solutions. Such effects decrease cell division, elongation and differentiation, leading to a decrease in final yield. As evident from our results, seed laser pretreatment significantly increased plant growth, photosynthetic pigments and nutrients ($Na^+$, $K^+$ and $Ca^{2+}$) status of water-stressed wheat plants resulting in a significant enhancement in grain yield [52,53]. A correlation between He–Ne laser irradiation and improvement in nutrient uptake has already been suggested by Qiu et al. in wheat [22].

## 5. Conclusions

Plant growth and yield contributing traits such as biomass accumulation, number of tillers, number of fertile tillers and 100 g grain weight were upregulated by seed laser priming, particularly under water stress. Further, laser seed priming also promoted photosynthetic pigments like chlorophyll *a* and *b*, total chlorophyll, carotenoids, and anthocyanin. The activities of antioxidative enzymes (SOD, POD, CAT and APX) were the key contributor to growth enhancement in plants grown from laser-primed seed. Then enhancement in total soluble proteins, total soluble sugars, total free amino acids and ions ($Na^+$, $K^+$ and $Ca^{2+}$) induced by laser-priming in water-stressed plants significantly modulated plant growth resulting in higher production of fertile tillers and 100-grain yield. However, some attributes were unregulated under 1 or 5 min laser exposure time, 2 min exposure time was the most effective. Cultivar Anaj-2017 had a better response to laser exposure than FSD-2008.

**Author Contributions:** Project administration, H.A.; supervision, M.S.A.A.; Conceptualization, A.K.A.; data curation, E.-S.A., W.R.; writing—review and editing, H.-u.-R.A., I.A.-A., K.F.A. and N.U. All authors have read and agreed to the published version of the manuscript.

**Funding:** This research was funded by Researchers Supporting Project number (RSP-2021/298), King Saud University, Riyadh, Saudi Arabia.

**Institutional Review Board Statement:** Not applicable.

**Informed Consent Statement:** Not applicable.

**Data Availability Statement:** The primary data, R codes and modeling details are available from Muhammad Sajid Aqeel Ahmad and can be requested if needed to reproduce the data visualization or other results.

**Acknowledgments:** The authors extend their appreciation to the Researchers Supporting Project number (RSP-2021/298), King Saud University, Riyadh, Saudi Arabia. This article has been extracted from the MPhil thesis of the Hamza Aslam submitted to University of Agriculture, Faisalabad, Pakistan.

**Conflicts of Interest:** No conflict of interest.

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
