# Peer review of "He–Ne Laser Priming Enhances Drought Tolerance in Wheat through Differential Modification of Photosynthetic Pigments and Antioxidative Enzymes"

_agronomy, doi:10.3390/agronomy12102376_

Round 1

Reviewer 1 Report

The topic of the presented publication, concerning the influence of laser priming on the physiological state of wheat plants subjected to water stress, is a current research problem, both from the point of view of science and agricultural practice. 

Yield is often determined by abiotic or biotic factors that reduce yield size and quality. Water deficit is one of the most important environmental stressors limiting plant growth and development. This stress causes direct or indirect disturbances in almost all physiological processes - water exchange, mineral nutrition, photosynthesis, growth, etc. The final result of the negative impacts is a reduced yield and a deterioration in produce quality.

In recent decades, significant progress has been made in researching the mechanisms of plant adaptation to water stress. Still, reliable methods for assessing this stressor and ways to reduce its negative influence on the physiological state of plants are being sought.

For this reason, I believe that studying this problem is useful for both science and practice.

The purpose of the work corresponds well to the title of the publication and the scope of the experiments conducted. The research methods are correctly selected and applied. They are described very accurately.

The results obtained are well documented and analyzed statistically. They are interconnected and allow reliable results to be obtained. I highly appreciate using Pearson's correlation matrices to interpret the results obtained.

            The literature provided is sufficient and properly selected.  

Author Response

Thanks for the valuable comments. There is no comment to address.

Reviewer 2 Report

The Manuscript Number # agronomy-1902611:  He-Ne laser priming enhances water stress tolerance of wheat through differential modification of photosynthetic pigments and activities of anti-oxidative enzymes” This research provides valuable information on the Effects of He-Ne laser priming enhancing water stress tolerance of wheat.

Major comments

1.      In the "Introduction," authors should include a brief about the recent studies using He-Ne laser priming to enhance water stress tolerance in different crops

2.      Authors should include results of Soil chemical composition in supplementary or inside the manuscript

3.      Discussion section …… Authors should discuss more He-Ne laser priming enhances water stress tolerance in different crops

4.      There are numerous grammatical and sentence problems in the manuscript. The author should polish the text with the aid of a native English editor.

Some Minor comments

1.      Line 76: respond to waters stress…….correct…. with water stress

2.      Line 243: recorded at 2 and 5 min. laser exposure ……….corrected with… recorded at 2 and 5 min. of laser exposure

3.      Line 421: seedlings………correct with seeding

4.      Line 425: showed that increase in time duration….. correct sentence  with Showed that an increase in the time duration

5.      Line 431:  wheat plants as observed in our study………..Correct with wheat plants, as observed in our study

6.      Line 435: This receptor has well known regulatory effects many plant…correct with effects on  many plants

Author Response

Major comments

  1. In the "Introduction," authors should include a brief about the recent studies using He-Ne laser priming to enhance water stress tolerance in different crops

Response: Added

  1. Authors should include results of Soil chemical composition in supplementary or inside the manuscript

Response: The soil used in this experiment was obtained from agricultural filed. The soil physicochemical composition was not determined and the data is not available for inclusion in this manuscript

  1. Discussion section …… Authors should discuss more He-Ne laser priming enhances water stress tolerance in different crops

Response: Added

  1. There are numerous grammatical and sentence problems in the manuscript. The author should polish the text with the aid of a native English editor.

Response: The manuscript has been read by an expert for English grammar and language

 Some Minor comments

  1. Line 76: respond to waters stress…….correct…. with water stress

Response: Corrected

  1. Line 243: recorded at 2 and 5 min. laser exposure ……….corrected with… recorded at 2 and 5 min. of laser exposure

Response: Corrected

  1. Line 421: seedlings………correct with seeding

Response: Corrected

  1. Line 425: showed that increase in time duration….. correct sentence  with Showed that an increase in the time duration

Response: Corrected

  1. Line 431:  wheat plants as observed in our study………..Correct with wheat plants, as observed in our study

Response: Corrected

  1. Line 435: This receptor has well known regulatory effects many plant…correct with effects on  many plants

Response: Corrected